# SADUNs: Sharpness-Aware Deep Unfolding Networks for Image Restoration

## Abstract

The ability to improve model performance while preserving structural integrity represents a fundamental challenge in deep unfolding networks (DUNs), particularly when handling increasingly complex black-box priors. This paper presents a novel Sharpness-Aware Deep Unfolding Networks (SADUNs), which addresses these limitations by integrating Sharpness-Aware Minimization (SAM) principles with the proximal operator theory. By analyzing the gradient landscape of linear inverse problems, we develop the separable sharpness-aware perturbation and subgradient calculation modules that maintain original network structures while enhancing optimization. Our theoretical analysis demonstrates that SADUNs achieve linear convergence for sparse coding tasks under common assumptions. Crucially, our framework reduces training costs through fine-tuning compatibility and preserves inference speed by eliminating redundant gradient computations via proximal operator properties. Comprehensive experiments validate SADUNs across multiple domains. Moreover, we have validated the improvement of our framework on plug-and-play single image super-resolution tasks, which means that our framework has the potential to expand to more types of deep unfolding networks.

## 1 Introduction

Linear Inverse Problems (LIPs) are a core research direction in science and engineering, focusing on inferring input information or system characteristics from observable outputs. Unlike well-posed forward problems, LIPs are typically ill-posed but indispensable in practical scenarios like medical imaging (Sun et al., 2016) and signal processing (Zheng et al., 2022a). A major breakthrough in LIPs is compressive sensing (CS), which integrates signal acquisition and reconstruction efficiently. By exploiting signal sparsity (Baraniuk et al., 2010), CS enables sub-Nyquist-rate measurements, and original signals can be reconstructed from limited observations via optimization algorithms, finding wide use in image restoration (Cheng et al., 2022) and HSI (Zhang et al., 2022c).

CS is often modeled as the $l_1$-norm regularized Least Absolute Shrinkage and Selection Operator (LASSO) problem (to promote sparsity), with solutions including proximal gradient algorithms like Iterative Shrinkage-Thresholding Algorithm (ISTA) (Daubechies et al., 2004) and its variants (e.g., momentum-enhanced versions (Beck & Teboulle, 2009)). With deep learning advances, studies (e.g., (Gregor & LeCun, 2010)) accelerated such iterative algorithms by learning: unfolding ISTA iterations into "Learned ISTA (LISTA)" layers (like time-unfolded recurrent networks), forming the class of Deep Unfolding Networks (DUNs).

Among all DUNs, we can simply classify them into interpretability-oriented, application-oriented, and framework-oriented algorithms. As DUNs are designed from traditional iterative algorithms, some previous works such as LISTA-CP (Chen et al., 2018), focus on **interpretability** with sparsity-based priors. However, in real world **applications**, people are not satisfied with the $l_1$-norm, as it's a convex approximation of $l_0$-norm. As conventional optimization employs non-convex regularizers (Fan & Li, 2001), deep learning admits black-box priors (Zhang & Ghanem, 2018; You et al., 2021; Wang & Gan, 2024; Zhang et al., 2022c; Yang et al., 2025). Moreover, the neural-network modules corresponding to these black-box priors grow increasingly complexity. The works (Zheng et al., 2022b) and (Li et al., 2021) have proposed acceleration **frameworks** of HNO and ELISTA for unfolding networks, respectively. HLISTA designed a framework that embeds complex neural networks into simple DUNs to enhance performance.

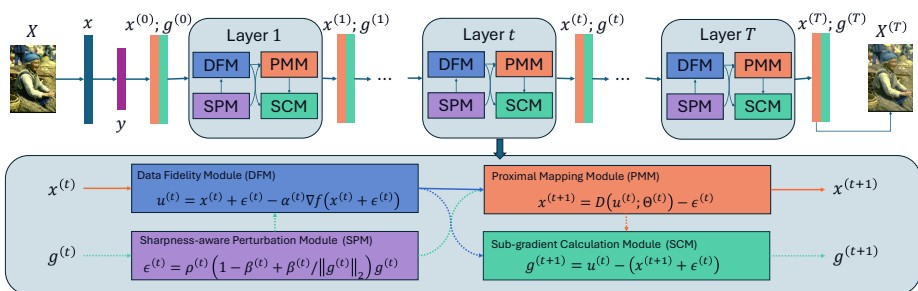

Figure 1: Illustration of our proposed SADUNs framework. Specifically, SADUNs unfolds $T$ iterations to learnable layers, $D$ is parameterized proximal mapping, and we use the update rule of Unified Sharpness-Aware Minimization in SPM. We use dotted arrows to indicate our modifications to the traditional DUN model. When these connections fail (just set $\rho = 0$), our model will degenerate to the traditional DUN. In other words, to convert a traditional DUN to a SADUN, just restore these connections. This may help you better understand our tuning strategy.

Inspired by TV-layers (Yeh et al., 2022), we realize that research on DUNs in recent years has often focused on designing better modules or introducing more complex algorithms, and these methods have not utilized the common smoothing and sharpening techniques in the field of image processing. Recently, the emergence of Sharpness-Aware Minimization (SAM) (Zhou et al., 2021) in deep learning has rekindled interest in loss-landscape geometry with only first-order information, which means that we can indirectly utilize part of the sharpness information by introducing the SAM algorithm. Furthermore, DUNs are widely applied to solving inverse problems with black-box priors. This means that we can use the SAM algorithm to extract the loss-landscape geometry information contained therein. Finally, when the existing framework-oriented methods were proposed, the network structures and parameter counts of DUNs were relatively simple, and their improved models were often trained in an end-to-end manner. However, more advanced DUNs usually have publicly available model files, so we aim to design a method that performs fine-tuning based on pre-trained models.

To address the aforementioned challenges, we design a deep unfolding framework based on the well-known SAM algorithm (Zhou et al., 2021), denoted as Sharpness-Aware Deep Unfolding Networks (SADUNs). We design separable sharpness-aware and subgradient calculation modules, which significantly reduce damage to the model, as depicted in Figure 1. The main contributions are summarized as follows:

- **A novel perspective and comprehensive framework for DUNs.** We commence from the gradient landscape of linear inverse problems and explore enhancing model performance by improving local problem properties, which offers a fresh perspective for designing more sophisticated DUNs. From the sharpness-aware perspective, we engineered a framework applicable to most deep unfolding networks (DUNs). By leveraging proximal operators and subgradients, we eliminate one gradient computation in sharpness-aware perturbation updates, resulting in virtually no inference speed degradation.

- **Linear convergence for sparse coding.** The theoretical results demonstrate that our network achieves linear convergence, which guarantees the applicability of our framework to scenarios demanding sparse-based priors, such as group-sparsity (Zou et al., 2024), low-rank (Ke et al., 2021).

- **Reduce training costs.** By emphasizing local properties, our framework inherently supports fine-tuning techniques akin to those in LLMs, enabling seamless migration from conventional DUNs to our SADUNs. Prior frameworks typically disregard complex priors (and their neural representations), thus heavily relying on end-to-end training.

- **Performance improvement for a variety of experiments.** We conduct extensive experiments, including synthetic data experiments, natural image compressive sensing and single image super resolution. The results show that our SADUNs architecture can effectively improve the performance of original networks and is widely applicable to different DUNs.

## 2 BACKGROUND AND PRELIMINARIES

### 2.1 ITERATIVE SHRINKAGE-THRESHOLDING ALGORITHM (ISTA)

For the LASSO problem mentioned above, which is used to model compressed sensing, its form is as follows:

$$\min_x \left\{ F(x) = f(x) + \lambda g(x) = \frac{1}{2}\|y - Ax\|_2^2 + \lambda\|x\|_1 \right\}, \tag{1}$$

where $y \in \mathbb{R}^m$ denotes the observed measurement vector, $A \in \mathbb{R}^{m \times n}$ (with $m \ll n$) represents the sensing matrix, $x \in \mathbb{R}^n$ is the original sparse signal to be reconstructed, and $\lambda > 0$ is a regularization parameter balancing the data-fitting term $f$ and regularization term $g$ to promote sparsity.

As one of the commonly used algorithms for solving LASSO, ISTA can be introduced by adopting the idea of majorize-minimization (MM) optimization (Ortega & Rheinboldt, 2000), which works by finding a surrogate function that minimizes the objective function.

**Definition 1** (Surrogate function). *In majorize-minimization, a surrogate function $Q(x \mid x^{(t)})$ is defined as a function that majorizes the original objective function $f(x)$ at the current iterate $x^{(t)}$, satisfying two key conditions:*

$$Q(x \mid x^{(t)}) \leq f(x), \forall x \in \mathbf{dom}\{f\}; Q(x^{(t)} \mid x^{(t)}) = f(x^{(t)}).$$

*This ensures that minimizing the surrogate function $Q(x \mid x^{(t)})$ to obtain the next iterate $x^{(t+1)}$ will be non-increasing in the original objective function $f$.*

A common choice of the surrogate function is obtained by performing a second-order Taylor expansion on $f$:

$$Q(x \mid z) = f(z) + (x - z)^\top \nabla f(z) + \frac{L}{2}\|x - z\|_2^2, \tag{2}$$

where $L$ is greater than the upper bound of the eigenvalues of $\nabla^2 f(z)$ and $z$ is a known point (usually $x^{(t)}$). In compressed sensing problems, this can be directly written as the upper bound of the eigenvalues of $A^\top A$. Then, we have

$$x^{(t+1)} = \underset{x}{\operatorname{argmin}} Q(x \mid x^{(t)}) + \lambda g(x) = \underset{x}{\operatorname{argmin}} \frac{L}{2}\|x - (x^{(t)} - \frac{1}{L}\nabla f(x^{(t)}))\|_2^2 + \lambda g(x),$$

with the following definition:

**Definition 2** (Proximal mapping/operator). *For any $x \in \mathbb{R}^n$, the proximal operator $\operatorname{prox}_{\lambda g}$ is the unique solution to the optimization problem:*

$$\operatorname{prox}_{\lambda g}(y) = \underset{x}{\operatorname{argmin}} \lambda g(x) + \frac{1}{2}\|x - y\|_2^2, \tag{3}$$

*where $g$ is a proper convex lower semi-continuous function, $\lambda > 0$ is a positive parameter, $\|\cdot\|_2$ denotes the Euclidean norm.*

Then we have

$$x^{(t+1)} = \operatorname{prox}_{\lambda/Lg}(x^{(t)} - \frac{1}{L}A^\top(Ax^{(t)} - y)), \tag{4}$$

where $\operatorname{prox}_{\lambda/Lg}$ is the soft-thresholding function $\eta_{\lambda/L}(x) = \operatorname{sgn}(x)\max\{|x| - \lambda/L, 0\}$ for the LASSO problem (1).

### 2.2 ISTA-BASED DUNs

(Gregor & LeCun, 2010) firstly proposed a class of methods to learn the parameters of the algorithm from training data, called deep unfolding networks (DUNs), and proposed a Learned ISTA (LISTA) method for the sparse coding task. Subsequently, by mining the relationships between variables, (Chen et al., 2018) provided the first linear convergence for DUNand presented the LISTA-CP method, whose update rule can be formulated as follows:

$$x^{(t+1)} = \eta_{\theta^{(t)}}(x^{(t)} - W^{(t)}(Ax^{(t)} - y)), \tag{5}$$

where the sequence of learnable parameters $\left\{W^{(t)}, \theta^{(t)}\right\}_{t=1}^{T}$ is initialized with $\alpha A^{\top}$ and $\alpha\lambda$, respectively, and $T$ represents the total number of iterations (or layers). In recent years, a large number of deep unfolding networks have emerged with clear convergence guarantees, such as (Wu et al., 2020; Li et al., 2021; Kong et al., 2022; Liu et al., 2018).

In order to achieve better sparse representation, (Zhang & Ghanem, 2018) proposed a method by using neural networks to promote sparsity, called ISTA-NET, which firstly introduces conventional layers to DUNs. By introducing deep models into the regularization term, unfolding networks have rapidly spread to various application fields, including natural image processing (Zhang & Ghanem, 2018; Wang & Gan, 2024), communication technology (Zheng et al., 2022a), and medical image processing (Zhang & Ghanem, 2018), and other areas (Han et al., 2020; Zhang et al., 2022a). For ISTA-based unfolding networks, the regularization term can usually be understood as a hidden function with parameters, i.e. $g(x, \Theta)$, where $\Theta$ is the set of learnable parameters in the proximal operator of $g(x, \Theta)$.

## 2.3 PROXIMAL OPERATORS AND SUBGRADIENTS

In Definition 2, we have already given the definition of the proximal operator. Here, we give the definition of the subgradient.

**Definition 3.** *For a convex function $f : \mathbb{R}^n \to \mathbb{R}$, a vector $v \in \mathbb{R}^n$ is called a subgradient of $f$ at a point $x \in \mathbb{R}^n$ if for all $y \in \mathbb{R}^n$, the following inequality holds:*

$$f(y) \geq f(x) + \langle v, y-x \rangle,$$

*where $\langle \cdot, \cdot \rangle$ denotes the inner product in $\mathbb{R}^n$. The set of all subgradients of $f$ at $x$ is called the subdifferential of $f$ at $x$, denoted by $\partial f(x)$:*

$$\partial f(x) = \left\{ v \in \mathbb{R}^n \mid f(y) \geq f(x) + \langle v, y-x \rangle, \forall y \in \mathbb{R}^n \right\}.$$

Next, we present two useful properties of the proximal operator and subgradients (Beck, 2017):

**Property 1.** *If $f(x) = g(ax + b)$ with $a > 0$, then*

$$\mathrm{prox}_{a^2 \lambda g}(ax + b) = a(\mathrm{prox}_{\lambda f}(x) + b). \tag{6}$$

**Property 2.** *According to definitions 2 and 3, for any proper convex lower semi-continuouswith function $g$, with $x^\star = \mathrm{prox}_{\lambda g}(x)$, we define*

$$\tilde{\nabla} g(x^\star) = x - x^\star \in \lambda \partial g(x^\star). \tag{7}$$

## 2.4 SHARPNESS-AWARE MINIMIZATION

The Sharpness-Aware Minimization (Foret et al., 2020) aims to improve the sharpness of the loss function by solving such minimax problems:

$$\min_x \max_{\|\epsilon\|_p \leq \rho} F(x + \epsilon), \tag{8}$$

where $F$ here means loss function in deep learning, $\rho$ represents the radius of the exploration area. (Andriushchenko & Flammarion, 2022; Si & Yun, 2023; Su et al., 2025) suggest that the perturbation is not required to be normalized, named Unnormalized Sharpness-Aware Minimization (USAM). Then (Oikonomou & Loizou, 2025) proposed a framework balanced between SAM and USAM:

$$\epsilon(x) = \rho(1 - \beta + \frac{\beta}{\|\nabla F(x)\|_2})\nabla F(x), \tag{9}$$

where $\beta \in [0, 1]$, called Unified SAM, which offers a single, theoretically grounded framework that generalizes and improves both SAM and USAM by relaxing restrictive assumptions, supporting arbitrary sampling strategies, and delivering SOTA convergence guarantees for nonconvex and PL functions. In particular, when $\beta$ takes the values of 0 and 1, respectively, Eq. (9) corresponds to SAM and USAM. Although some studies have focused on introducing adaptive gradients (Sun et al., 2024), their update to $x$ can still be expressed as:

$$x^{(t+1)} = x^{(t)} - \alpha g^{(t)}, \tag{10}$$

Table 1: Changed PSNR(dB) Results for Compressive Sensing on Set11.

| | house | came | lena | fing | mona | flin | parr | boat | fore | barb | pepp |
|---|---|---|---|---|---|---|---|---|---|---|---|
| sharpen | 0.009 | -0.004 | 0.002 | 0.016 | -0.004 | 0.011 | -0.000 | 0.002 | 0.000 | -0.001 | -0.011 |
| smooth | -0.033 | 0.003 | -0.014 | -0.047 | -0.007 | -0.039 | -0.006 | -0.015 | -0.013 | -0.003 | 0.012 |

where $g^{(t)}$ is the perturbation gradient $\nabla F(x^{(t)} + \epsilon(x^{(t)}))$ or its variants. The recently proposed SAM methods can be mainly divided into three categories: optimizing the perturbation direction (Zhou et al., 2021; Becker et al., 2024), optimizing the perturbation radius (Oikonomou & Loizou, 2025; Kwon et al., 2021), exploring better perturbation gradients (Sun et al., 2024). Note that, some variants of SAM (Sun et al., 2024; Mordido et al., 2023) tailored for stochastic optimization are incompatible with DUNs.

### 2.5 DUNs Lack Sharpness Information

To verify the neglect of sharpness information by DUNs (Deep Unrolling Networks), we applied a certain degree of sharpening and smoothing to the outputs of ISTA-NET respectively, and statistically analyzed the changes in PSNR with a 25% compression ratio, as shown in Table 1, which indicates that existing DUNs have certain limitations in capturing sharpness information.

In this experiment, we used the "gaussian" and "laplace" functions from scikit-image to smooth or sharpen the images:

$$x_{smooth} = \text{gaussian}(x, 0.3); x_{sharpen} = x + 0.01\text{laplace}(x).$$

## 3 Our Sharpness-Aware Minimization Architecture for ISTA-based Deep Unfolding Networks

Before introducing our algorithm, please note that for simplicity in the formula, we use $g(x)$ and $D(x)$ as a simplification of $g(x, \Theta)$ and $D(x, \Theta)$.

Unlike previous frameworks, which are designed to solve the problem (1), our framework focuses on its gradient landscape, by solving the problem:

$$x^{(t+1)} = \underset{x}{\operatorname{argmin}} f(x + \epsilon^{(t)}) + \lambda g(x + \epsilon^{(t)}), \tag{11}$$

where the perturbation $\epsilon^{(t)}$ is defined as:

$$\epsilon^{(t)} = \underset{\|\epsilon\|_2 \leq \rho}{\operatorname{argmax}} f(x^{(t)} + \epsilon) + \lambda g(x^{(t)} + \epsilon). \tag{12}$$

### 3.1 Solve Problem (11) with Property 1 and Majorize-Minimization.

Looking back at ISTA and MM optimization, we first provide the definition of the surrogate function $Q(x + \epsilon^{(t)} \mid z^{(t)})$ for Eq. (11) as follows:

$$f(z^{(t)}) + (x + \epsilon^{(t)} - z^{(t)})^\top \nabla f(z^{(t)}) + \frac{1}{2\alpha^{(t)}} \|x + \epsilon^{(t)} - z^{(t)}\|_2^2,$$

where $\alpha^{(t)} \leq 1/L$ is the step size of $t$-th iteration. Thus, according to the MM optimization criterion, we use $\tilde{Q}(x + \epsilon \mid z^{(t)})$ to replace $f(x + \epsilon^{(t)})$, resulting in the following:

$$x^{(t+1)} = \underset{x}{\operatorname{argmin}} \tilde{Q}(x + \epsilon^{(t)} \mid z^{(t)}) + \lambda g(x + \epsilon^{(t)}). \tag{13}$$

Next, we will explicitly solve for Eq. (13). Recalling Property 1, we need to construct $v$, such that $v(x) = g(x + \epsilon^{(t)})$, that is:

$$
\begin{aligned}
x^{(t+1)} &= \operatorname{prox}_{\alpha^{(t)}\lambda v}(x^{(t)} - \alpha^{(t)}\nabla f(z^{(t)})) \\
&= \underset{x}{\operatorname{argmin}} \tilde{Q}(x + \epsilon^{(t)} \mid x^{(t)} + \epsilon^{(t)}) + \lambda v(x),
\end{aligned}
$$

Then, we have: $\text{prox}_{\lambda g}(z^{(t)}) = \text{prox}_{\lambda u}(x^{(t)}) + \epsilon^{(t)}$, where $\text{prox}_{\lambda g}(x) = D(x - \alpha^{(t)}\nabla f(x))$. Thus, we obtain the iterative form corresponding to Eq. (13):

$$x^{(t+1)} = \text{prox}_{\alpha^{(t)}\lambda g}(z^{(t)} - \alpha^{(t)}\nabla f(z^{(t)})) - \epsilon^{(t)}, \tag{14}$$

which is similar to (10), since (10) is actually equivalent to:

$$x^{(t+1)} = (x^{(t)} + \epsilon^{(t)}) - \nabla F(x^{(t)} + \epsilon^{(t)}) - \epsilon^{(t)}.$$

## 3.2 CALCULATING SUBGRADIENT WITH PROPERTY 2.

For the perturbation sub-problem (12), we continue to use the update strategy of unified SAM, namely:

$$\epsilon^{(t)} = \rho^{(t)}(1 - \beta^{(t)} + \frac{\beta^{(t)}}{g^{(t)}})g^{(t)} \tag{15}$$

where $g^{(t)} \in \partial F(x^{(t)})$. However, for ISTA-NET or other complex DUNs, the subgradient of the regulation term $\partial g(x)$ is not readily available. Thus, we attempt to estimate the subgradient of $v$ at $x^{(t+1)}$ by:

$$\alpha^{(t)}\lambda\tilde{\nabla}v(x^{(t+1)}) = x^{(t)} - \alpha^{(t)}\nabla f(z^{(t)}) - x^{(t+1)}.$$

However, we need the subgradient of $g$ rather than $v$. According to $v(x) = g(x + \epsilon^{(t)})$, we can obtain

$$\alpha^{(t)}\lambda\tilde{\nabla}g(x^{(t+1)} + \epsilon^{(t)}) = x^{(t)} - \alpha^{(t)}\nabla f(z^{(t)}) - x^{(t+1)}.$$

Fortunately, SAM allows for certain variations in the selection of gradients when calculating perturbations (Zhou et al., 2021; Du et al., 2021). Thus, we present the update formula for perturbations:

$$\epsilon^{(t)} = \rho(1 - \beta + \frac{\beta}{\|\tilde{\nabla}g(x^{(t)} + \epsilon^{(t-1)})\|_2})\tilde{\nabla}g(x^{(t)} + \epsilon^{(t-1)}). \tag{16}$$

## 3.3 A SUMMARY OF OUR SADUNs FRAMEWORK

To better illustrate our model, we decompose it into four components. First, there are two modules corresponding to the deep unfolding network: the Data Fidelity Module (DFM), which is derived from the Taylor expansion of the data fidelity term, and the Proximal Mapping Module (PMM), which enforces the solution to satisfy the prior knowledge. Additionally, the two modules dedicated to sharpness awareness include the Sharpness-Aware Perturbation Module (SPM) and the Subgradient Calculation Module (SCM). The overall structure of our framework is depicted in Algorithm 1. Figure 1 more intuitively illustrates our model, where solid lines represent the data flow of DUNs, and dashed lines denote the interaction between DUNs and SAM.

---

**Algorithm 1** SADUNs

---

**Input:** Observation $y$, basis matrix $A$, depth $T$, scalar parameters $\left\{\alpha^{(t)}, \beta^{(t)}, \Theta^{(t)}, \rho^{(t)}\right\}_{t=1}^{T}$, initial point $x^{(0)} = 0$ and initial gradient $g^{(0)} = 0$
**for** $t = 0$ **to** $T - 1$ **do**
    SPM: $\epsilon^{(t)} = \rho^{(t)}(1 - \beta^{(t)} + \frac{\beta^{(t)}}{\|g^{(t)}\|})g^{(t)}$;
    DFM: $u^{(t)} = x^{(t)} + \epsilon^{(t)} - \alpha^{(t)}\nabla f(x^{(t)} + \epsilon^{(t)})$;
    PMM: $x^{(t+1)} = D(u^{(t)} + \epsilon^{(t)}, \Theta^{(t)}) - \epsilon^{(t)}$;
    SCM: $g^{(t+1)} = x^{(t+1)} + \epsilon^{(t)} - u^{(t+1)}$;
**end for**

---

## 3.4 LEARNING STRATEGY

By exploring proximal operator properties, each module in the original model has a direct counterpart in SADUN. In other words, we can simply regard the original DUN as special SADUN with $\rho = 0$, which means we can reuse the well-trained model. Therefore, one may load trained parameters of the original DUN and initialize $\rho$ and $\beta$. Then, you may fine tune SADUN for a few epochs or perform

grid searches on rho and beta to avoid any training. In the experimental section, we used end-to-end training in sparse coding tasks, fine tune strategy in compressive sensing tasks, and no training in the final plug-and-play experiment. Our framework yields consistent improvements across these disparate training strategies.

## 4 THEORETICAL RESULTS

Since our framework (i.e. Algorithm 1) can be adapted to LISTAs, we prove that our framework can maintain linear convergence under sparse prior conditions in this section. Firstly, we introduce some definitions and assumptions from (Chen et al., 2018; Liu et al., 2018). Due to the introduction of sparsity priors, we make the following assumptions about the set of sparse vectors.

**Assumption 1** (Basic Assumption). *Sparse signal $x^\star$ is sampled from the following set:*

$$x^\star \in \{x^\star \mid |x_i^\star| \le B, \forall i, \|x\|_0 \le s\}. \tag{17}$$

*In other words, $x^\star$ is bounded and $s$-sparse ($s \ge 2$).*

Note that this assumption is a basic assumption for sparse coding. To my knowledge, almost all LISTAs need to satisfy this assumption. In addition, the matrix $W^{(t)}$ learned in (1) must meet the following definition.

**Definition 4.** *For given $A \in \mathbb{R}^{m \times n}$, the generalized mutual coherence is defined as*

$$\mu(\mathbf{A}) = \inf_{\substack{\mathbf{W} \in \mathbb{R}^{N \times M} \\ \mathbf{W}_i^T \mathbf{A}_i = 1, 1 \le i \le M}} \left\{ \max_{\substack{i \ne j \\ 1 \le i,j \le M}} \mathbf{W}_i^T \mathbf{A}_j \right\}. \tag{18}$$

*Additionally, We define $W(A)$ as the set of $W$ which attains infimum given (18). A weight matrix $W$ is "good" if*

$$W \in \left\{ W \mid |W_i^\top A_j| \le \mu(A) \forall j \ne i, W_i^\top A_i = 1, \forall i \right\}$$

From Lemma 1 in (Chen et al., 2018), we know $W(A) \ne \emptyset$. Furthermore, the lower bound of thresholding $\theta^{(t)}$ should be given to make $x^{(t+1)}$ satisfies *No False Positives*. Then, we have the following theorem.

**Theorem 1.** *Given $\left\{W^{(t)}, \theta^{(t)}\right\}_{t=0}^{\infty}$ and $x^{(0)} = 0$, let $\left\{x^{(t)}\right\}_{t=0}^{\infty}$ be generated by Algorithm 1. If Assumption 1 holds and $s$ is sufficiently small, then there exists a sequence of parameters $\left\{W^{(t)}, \theta^{(t)}\right\}_{t=0}^{\infty}$ such that, for all $x^\star \in \mathcal{X}(Bs)$, we have*

$$\|x^{(t)}(x^\star) - x^\star\|_2 \le sB \exp(-ct),$$

*where $c > 0$ is related to $A$, $s$ and sufficiently small $\rho$ for all $\beta \in [0, 1]$.*

## 5 EXPERIMENTS

To verify the generality of the improvements brought by our framework to deep unrolling networks, we conduct experiments on three types of proximal operators: those using the soft-thresholding function (LASSO), those relying solely on convolutional operations (ISTA-NET), and those adopting state-of-the-art mechanisms such as CNV2 and multi-head attention (UFC-NET). All experiments are performed on a server with NVIDIA 2080 Ti.

### 5.1 SYNTHETIC DATA SPARSE CODING (LASSO)

To verify the effectiveness of our Theorem 1, we conducted sparse representation experiments on the LASSO model on synthetic data. We adopted our SADUNs framework to LISTA-CP, LISTA-CPSS(Chen et al., 2018), Analysis LISTA(Liu et al., 2018), named SALISTA-CP, SALISTA-CPSS, SALISTA-ANA. And we compared those algorithms with three noise levels expressed by SNR (Signal-to-Noise Ratio), which is the indicator and condition numbers $\kappa$ of ill conditioned matrix on sparse coding problems. We will use the same experimental setup as (Liu et al., 2018), with

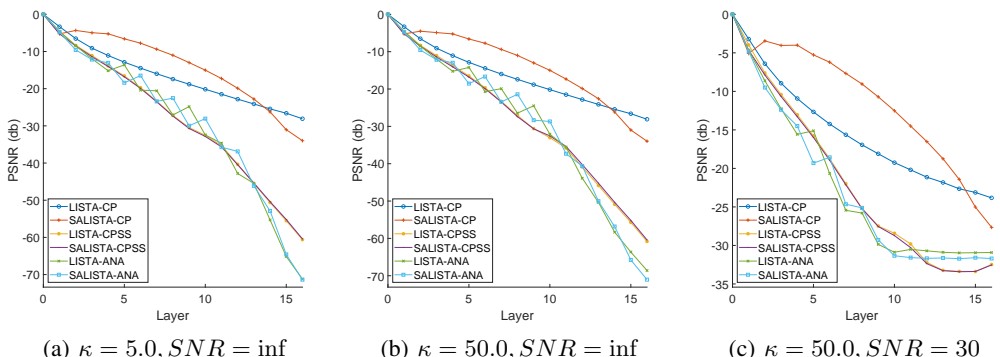

|  (a) $\kappa = 5.0, SNR = \inf$ | (b) $\kappa = 50.0, SNR = \inf$ | (c) $\kappa = 50.0, SNR = 30$ |

Figure 2: Comparisons of sparse representation with different layers under different SNR and $\kappa$.

$m = 250, n = 100$, and $T = 16$. All the results are shown in Figure 2, where NMSE is defined as following:

$$\text{NMSE}(x, x^\star) = 10 \log_{10} \left( \frac{\mathbb{E}\|x - x^\star\|_2^2}{\mathbb{E}\|x^\star\|_2^2} \right), \tag{19}$$

where $x$ represents the output of the networks. Our framework demonstrates substantial improvements over generic DUNs (e.g., LISTA-CP), while still offering noticeable gains for inherently stronger models (e.g., LISTA-CPSS).

## 5.2 NATURAL IMAGE COMPREHENSIVE SENSING

To better demonstrate the applicability of our algorithm to different unfolding networks, we have improved both the classic ISTA-NET and the SOTA UFC-NET by adopting our framework.

### 5.2.1 ISTA-NET

In this subsection, we perform a natural image compressive sensing task to evaluate ours and many other methods. We use the training set, sampling matrix, and initialization matrix provided by ISTA-NET, and tune our model using the same strategy. The proximal operator is defined as $D(u, \Theta) = \tilde{\mathcal{F}}(\eta_\theta(\mathcal{F}(u, \Theta_1)), \Theta_2)$, where $\Theta = \{\Theta_1, \theta, \Theta_2\}$ And, the recovery transform $\tilde{\mathcal{F}}$ satisfying the symmetry constrain $\tilde{\mathcal{F}} \odot \mathcal{F} = \mathcal{I}$, where $\mathcal{I}$ represent the identity mapping.

The results with different CS ratios are reported in Table 2, compared with TVAL3 (Li et al., 2013), D-AMP (Metzler et al., 2016), IRCNN (Zhang et al., 2017), SDA (Mousavi et al., 2015) and ReconNet (Kulkarni et al., 2016b). From all the results, we know that our SADUN-ISTA-NET architecture can effectively improve the performance of ISTA-NET. Moreover, our SADUN-ISTA-NET outperforms the other methods. And, we conducted visual comparisons of images at a 25% compression ratio and

Table 2: Comparisons of average PSNR (dB) performance on Set11 (Kulkarni et al., 2016a) with different CS ratios.

| Algorithms | CS Ratio (%) | | | | | | | Time Cost | |
|---|---|---|---|---|---|---|---|---|---|
| | 1 | 4 | 10 | 25 | 30 | 40 | 50 | CPU | GPU |
| TVAL3 | 16.43 | 18.75 | 22.99 | 27.92 | 29.23 | 31.46 | 33.55 | 3.135s | - |
| D-AMP | 5.21 | 18.40 | 22.64 | 28.46 | 30.39 | 33.56 | 35.92 | 51.21s | - |
| IRCNN | 7.70 | 17.56 | 24.02 | 30.07 | 31.18 | 34.06 | 36.23 | - | 68.42s |
| SDA | 17.29 | 20.12 | 22.65 | 25.34 | 26.63 | 27.79 | 28.95 | - | 0.0032s |
| ReconNet | 17.27 | 20.63 | 24.28 | 25.60 | 28.74 | 30.58 | 31.50 | - | 0.016s |
| ISTA-NET | **17.45** | 21.38 | 26.11 | 30.80 | 33.29 | 35.49 | 37.46 | 0.024s | 0.0036s |
| Ours | 17.40 | **21.46** | **26.18** | **31.96** | **33.34** | **35.63** | **37.57** | 0.025s | 0.0059s |

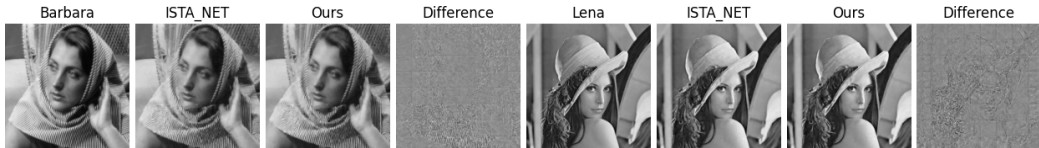

Figure 3: Visual comparisons between ISTA-NET and Ours SAISTA-NET

Table 3: Average PSNR(dB) Results for Compressive Sensing on the CBSD68 (Martin et al., 2002) Dataset.

| CS Ratio (%) | | 1 | 4 | 5 | 10 |
|---|---|---|---|---|---|
| UFC-NET | | 23.31 | 26.74 | 27.47 | 30.11 |
| SAUFC-T | $\beta = 1.0$ | 23.35 | 26.80 | 27.58 | 30.21 |
| | $\beta = 0.5$ | 23.34 | 26.81 | 27.60 | 30.22 |
| | $\beta = 0.0$ | 23.33 | 26.80 | 27.58 | 30.21 |
| SAUFC-F | $\beta = 1.0$ | 23.34 | 26.81 | 27.57 | 30.22 |
| | $\beta = 0.5$ | 23.34 | 26.81 | 27.57 | 30.22 |
| | $\beta = 0.0$ | 23.34 | 26.81 | 27.57 | 30.22 |

calculated the differences between the output images of our algorithm and those of ISTA-NET, as shown in Figure 3.

### 5.2.2 UFC-NET

Since our framework being designed for complex priors, we designed the fundamental experiments on SOTA DUNs such as UFC-NET (Wang & Gan, 2024) compared to previous frameworks. The training details such as datasets, optimizers are the same as UFC-NET, and we use a fixed learning rate. The UFC-NET introduced advanced modules such as Multi-head Attention Residual Block (MARB) and Auxiliary Iterative Reconstruction Block (AIRB) to achieve SOTA performance. We compare the tuned model with ISTA-NET$^+$ (Zhang & Ghanem, 2018), MAC-NET (Chen et al., 2020), AMP-NET (Zhang et al., 2020), LTw-ISTA (Gan et al., 2023) and original UFC-NET, and the results are shown in Table 4. However, we can not confirm whether the success of our framework comes from adjusting the structure and parameters. Thu, we further try to use fixed $\rho, \beta$ as in SAUFC-F, and the results are shown in Table 3. And, in Table 4, our SAUFC-NET demonstrates nearly consistent improvements in the SSIM metric, particularly on the Set14 and General100 datasets, where our method also achieves gains in PSNR.

## 6 FURTHER THOUGHTS FOR PLUG-AND-PLAY PRIORS

For plug-and-play models (PnP-DUNs), $\text{prox}_{\lambda/\mu g}(x)$ is often regarded as a well-trained denoiser. In the section, we take single image super resolution (SISR) as an example to varify our SADUNs can be adopted to free-formed priors. The half-quadratic splitting (HQS) algorithm (Geman & Yang, 1995) is often used in PnP-DUNs (Tang et al., 2025; Zhang et al., 2022b; Sinha & Chaudhury, 2025; Sinha et al., 2025). In order to decouple the data term and prior term of (1), HQS introduces an auxiliary variable $z$, which reformulate Problem (1):

$$\min_{x,z} f(x) + \lambda g(z) + \frac{\mu}{2}\|x - z\|_2^2,$$

where $\mu$ is a penalty parameter. It is obvious that HQS transforms linear inverse problems into two-step proximal operations:

$$z^{(t+1)} = \text{prox}_{\lambda/\mu g}(\text{prox}_{1/\mu f}(z^{(t)})). \tag{20}$$

According to Definition 2, $\text{prox}_{1/\mu f}$ satisfies $0 \in x - z + \frac{1}{\mu}\partial f(x)$, where $x = \text{prox}_{\lambda f}(z)$. For LIPs, $f$ is strongly convex, which means $\partial f(x) = \{\nabla f(x)\}$. Combining (29) and (30), we derive:

$$x = z - \frac{1}{\mu}\nabla f(x). \tag{21}$$

Table 4: Average PSNR (dB) and SSIM comparisons of UFC-Net and competing methods on multiple datasets with different CS ratios.

| Datasets | | Set14 (Zeyde et al., 2012) | | | | Urban100 (Huang et al., 2015) | | | | General100 (Dong et al., 2016) | | | |
|---|---|---|---|---|---|---|---|---|---|---|---|---|---|
| CS Ratio (%) | | 1 | 4 | 10 | 25 | 1 | 4 | 10 | 25 | 1 | 4 | 10 | 25 |
| ISTA-NET+ | PSNR | 18.20 | 22.07 | 25.98 | 30.610 | 16.66 | 19.65 | 23.48 | 28.89 | 19.00 | 23.74 | 28.52 | 34.31 |
| | SSIM | 0.4012 | 0.5707 | 0.7288 | 0.8699 | 0.1450 | 0.6486 | 0.7841 | 0.8944 | 0.4698 | 0.6545 | 0.8100 | 00.9248 |
| MAC-NET | PSNR | 18.43 | 23.71 | 26.40 | 30.67 | 16.39 | 21.60 | 24.49 | 28.79 | 19.72 | 26.17 | 29.70 | 34.83 |
| | SSIM | 0.3974 | 0.6171 | 0.7381 | 0.8742 | 0.3637 | 0.6120 | 0.7465 | 0.8798 | 0.4857 | 0.7169 | 0.8275 | 0.9283 |
| AMP-NET | PSNR | 21.55 | 25.42 | 28.70 | 33.12 | 19.55 | 22.73 | 25.92 | 30.79 | 22.68 | 26.91 | 30.77 | 35.93 |
| | SSIM | 0.5301 | 0.6996 | 0.8179 | 0.9136 | 0.5016 | 0.6819 | 0.8144 | 0.9188 | 0.6109 | 0.7689 | 0.8712 | 0.9493 |
| LTw-IST | PSNR | 21.48 | 25.44 | 28.82 | 33.40 | 19.46 | 23.01 | 26.76 | 31.79 | 22.69 | 27.53 | 31.91 | 37.31 |
| | SSIM | 0.5190 | 0.7112 | 0.8342 | 0.9241 | 0.4886 | 0.7061 | 0.8463 | 0.9349 | 0.5989 | 0.7935 | 0.8990 | 0.9616 |
| UFC-NET | PSNR | **21.79** | 25.67 | 29.09 | 33.81 | **19.68** | 23.36 | **27.54** | **32.81** | **23.08** | **27.92** | **32.31** | 37.75 |
| | SSIM | **0.5323** | 0.7163 | 0.8362 | 0.9259 | **0.5039** | 0.7193 | **0.8581** | 0.9421 | 0.6145 | 0.7988 | 0.9014 | 0.9624 |
| SAUFC-NET | PSNR | 21.74 | **25.70** | **29.15** | **33.91** | 19.65 | **23.37** | 27.51 | **32.81** | 22.94 | 27.90 | **32.31** | **37.82** |
| | SSIM | **0.5323** | **0.7183** | **0.8377** | **0.9273** | 0.5014 | **0.7201** | 0.8575 | **0.9427** | **0.6147** | **0.8003** | **0.9021** | **0.9631** |

Table 5: Average PSNR (dB) Results of Different Methods for 2x Single Image Super-Resolution on the CBSD68 Dataset.

| kernel | 1 | 2 | 3 | 4 | 5 | 6 | 7 | 8 |
|---|---|---|---|---|---|---|---|---|
| DPIR-IRCNN | 33.77 | 33.84 | 30.80 | 27.25 | 28.21 | 27.48 | 27.31 | 26.75 |
| SADUN-IRCNN($\beta$=1.0) | 33.77 | 33.84 | 30.80 | **27.27** | **28.22** | **27.49** | **27.32** | **26.77** |
| SADUN-IRCNN($\beta$=0.5) | 33.77 | 33.84 | 30.80 | 27.26 | 28.21 | 27.49 | 27.32 | 26.76 |
| SADUN-IRCNN($\beta$=0.0) | 37.77 | 33.84 | 30.80 | 27.26 | 28.21 | 27.48 | 27.31 | 26.75 |

From the perspective of ordinary differential equations (An et al., 2022), ISTA (4) and HQS (20) are solutions to the same differential equation. This means that the subgradient calculated based on ISTA can be regarded as an approximation of the HQS global gradient.

## 6.1 SINGLE IMAGE SUPER RESOLUTION (HQS)

The mathematical formulation of classical degradation model is given by

$$y = (x * k) \downarrow_s + n, \tag{22}$$

where $\downarrow_s$ denotes the standard $s$-fold downsampler, i.e., selecting the upper-left pixel for each distinct $s \times s$ patch and $k$ denotes the blur kernel. The classical SISR model still belongs to the linear inverse problem. HQS updates the data fidelity term $f$ using a closed-form solution. We use the same setting with (Zhang et al., 2022b), and we use pretrained IRCNN. With $\rho = 0.01$, our SADUN-IRCNN makes a slight promotion without tuning, the results are shown in Table 5. That is to say, even without tuning, applying the approximate subgradient SAM to the HQS-based DUNs is still effective, and only requires very little additional computation.

## 7 CONCLUSION AND FUTURE WORKS

The sharpness-aware framework can achieve significant performance improvements with the addition of parameters that are much smaller than those of most unfolding networks. Since the change in the number of parameters is minimal and the meaning of each component remains unchanged, our framework does not require full end-to-end training and only needs tuning on existing models. This means that our framework has better adaptability to large models compared to existing unfolding network frameworks. For future research, there is hope to further improve methods, such as applying gradient landscape and subgradient based methods to more types of DUNs and introducing acceleration mechanisms.

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

## A APPENDIX

### A.1 PROOF FOR SALISTA-CP

Before proving Theorem 1, we give the formulation of SALISTA-CP as following:

$$
\begin{align}
\epsilon^{(t)} &= m^{(t)} \odot \rho^{(t)}(1 - \beta^{(t)} + \beta^{(t)}/\|g^{(t)}\|_2)g^{(t)}, \tag{23} \\
u^{(t)} &= x^{(t)} - W^{(t)}(A(x^{(t)} + \epsilon^{(t)}) - y), \tag{24} \\
x^{(k+1)} &= \eta_{\theta^{(t)}}(u^{(t)}) - \epsilon^{(t)}, \tag{25} \\
g^{(t+1)} &= u^{(t)} - x^{(t+1)}, \tag{26} \\
m^{(t+1)} &= 1_{u^{(t)} > \theta^{(t)}}, \tag{27}
\end{align}
$$

where $1_c$ represents indicator function of set $c$, and $\odot$ means element-wise multiplication. In fields where traditional priors such as sparsity are employed, Deep Unfolding Networks (DUNs) often need to learn more information from the data fidelity term $f$, and the update rule for $u^{(t)}$ (24) is derived from LISTA-CP (Chen et al., 2018). Moreover, since subgradient $g^{(t)}$ is not sparse, this violates the *no-false-positive* assumption. We adopt a strategy similar to SSAM (Mi et al., 2022), where a mask $m^{(t)}$ is applied to the perturbations to ensure the sparsity of the solution. In this proof, we use the notion $x^{(t)}$ to replace $x^{(t)}(x^\star)$ for simplicity. We fix $A$ in the proof, $\mu(D)$ can be simply written as $\mu$.

*Proof.* A.1.1  PROOF FOR SALISTA WITH $\beta = 0$

**Step 1: *No False Positives*.**
Let $S = \text{support}(x^\star)$ indicates the non-zero entires. We want to prove by induction that, as long as all trained $W^{(t)}$ satisfies the "good" conditions in Definition 4, $x_i^{(t)} = 0, i \notin S$ (*no false positives*). As we set $x^{(0)} = 0$, it is satisfied when $t = 0$ and

$$
\theta^{(t)} = (\mu + \mu\rho^{(t)}) \sup_{x^\star \in \mathcal{X}(B,s)} \{\|x^{(t)} - x^\star\|_1\} + \mu \sum_{v=1}^{t-1}(1 + \mu(s-1))^{t-v} \prod_{b=v+1}^{t} \rho^{(b)} \sup_{x^\star \in \mathcal{X}(B,s)} \{\|x^{(v)} - x^\star\|_1\},
$$

where $\mathcal{X}(B,s) = \{x^\star \mid |x_i^\star| \le B, \forall i, \|x\|_0 \le s\}$ is defined in the Basic Assumption 1. Fixing $t$ and assuming $x_i^{(v)} = 0, i \notin S, \forall v \in \mathbb{N}^+ \le t$, then we have

$$
\begin{align*}
x_i^{(t+1)} &= \eta_{\theta^{(t)}}(x_i^{(t)} + \epsilon_i^{(t)} - W_{i,:}^{(t)}(A(x^{(t)} + \epsilon^{(t)}) - y)) - \epsilon_i^{(t)} \\
&= \eta_{\theta^{(t)}}(-W_{i,:}^{(t)}(A(x^{(t)} + \epsilon^{(t)}) - y)), i \notin S,
\end{align*}
$$

where $\epsilon_i^{(t)} = 0$ as the mask in (27). Since $W^{(t)}$ is good,

$$
\begin{align}
\theta^{(t)} &\ge (\mu + \mu\rho^{(t)})\|x^{(t)} - x^\star\|_1 + \mu \sum_{v=1}^{t-1}(1 + \mu(s-1))^{t-v} \prod_{b=v+1}^{t} \rho^{(b)}\|x^{(v)} - x^\star\|_1 \notag \\
&\ge \mu(\|x^{(t)} - x^\star\|_1 + \|\epsilon_j^{(t)}\|_1) \notag \\
&\ge \sum_{j \in S}(|W_{i,:}^{(t)} A_{:j}(x_i^{(t)} - x_j^\star)| + |W_{i,:}^{(t)} A_{:j}\epsilon_j^{(t)}|) \tag{28} \\
&\ge \sum_{j \in S} |W_{i,:}^{(t)} A_{:j}(x_j^{(t)} + \epsilon_j^{(t)} - x_j^\star)|, \forall i \in S,
\end{align}
$$

where we can achieve (28) with the following recursive formula:

$$
\begin{aligned}
\|\epsilon^{(t)}\|_1 &= \rho^{(t)}\|m^{(t-1)} \odot g^{(t-1)}\|_1 \\
&\leq \rho^{(t)}\sum_{i\in S}|-\epsilon_i^{(t-1)}-(x_i^{(t)}-x_i^\star)-\sum_{j\neq i,j\in S}W_{i,:}^{(t-1)}A_{:,j}(x_j^{(t-1)}+\epsilon_j^{(t-1)}-x_j^\star)| \\
&\leq \rho^{(t)}(\|\epsilon^{(t-1)}\|_1+\|x^{(t)}-x^\star\|_1+\mu\sum_{i\in S}\sum_{j\neq i,j\in S}|(x_j^{(t-1)}+\epsilon_j^{(t-1)}-x_j^\star)|) \\
&\leq \rho^{(t)}(\|\epsilon^{(t-1)}\|_1+\|x^{(t)}-x^\star\|_1+\mu(s-1)(\|\epsilon^{(t-1)}\|_1+\|x^{(t-1)}-x^\star\|_1)) \\
&= \rho^{(t)}\|x^{(t)}-x^\star\|_1+\sum_{v=1}^{t-1}(1+\mu(s-1))^{t-v}\prod_{b=v+1}^{t}\rho^{(b)}\|x^{(v)}-x^\star\|_1.
\end{aligned}
\tag{29}
$$

For (29), the mask $m^{(t)}$ satisfies $m_i^{(t)}=1 \Rightarrow i \in S$.

**Step 2: Upper Bound of Recovery Error.**
$\forall i \in S$, we have

$$
\begin{aligned}
x_i^{(t+1)} &= \eta_{\theta^{(t)}}(x_i^{(t)}+\epsilon_i^{(t)}-W_{i,:}^{(t)}(A(x^{(t)}+\epsilon^{(t)})-y))-\epsilon_i^{(t)} \\
&= \eta_{\theta^{(t)}}(x_i^{(t)}+\epsilon_i^{(t)}-\sum_{j\in S,j\neq i}W_{i,:}^{(t)}A_{:,j}(x_j^{(t)}+\epsilon_j^{(t)}-x_j^\star)-(x_i^{(t)}+\epsilon_i^{(t)}-x_i^\star))-\epsilon_i^{(t)} \\
&= \eta_{\theta^{(t)}}(x_i^\star-\sum_{j\in S,j\neq i}W_{i,:}^{(t)}A_{:,j}(x_j^{(t)}+\epsilon_j^{(t)}-x_j^\star))-\epsilon_i^{(t)} \\
&\in x_i^\star-\epsilon_i^{(t)}-\sum_{j\in S,j\neq i}W_{i,:}^{(t)}A_{:,j}(x_j^{(t)}+\epsilon_j^{(t)}-x_j^\star)-\theta^{(t)}\partial g(x_i^{(t+1)}+\epsilon_i^{(t)})
\end{aligned}
$$

where $\partial g$ denotes the sub-gradient of $\|\cdot\|_1$ that is defined by

$$
\partial g(x)=\begin{cases}\operatorname{sgn}(x), & x\neq 0, \\ [1,-1], & x=0.\end{cases}
\tag{30}
$$

Equation 30 suggests that $g(x_i^{(t+1)}+\epsilon_i^{(t)})$ has a magnitude not greater than 1. Thus, we obtain for $i \in S$,

$$
\begin{aligned}
|x_i^{(t+1)}-x_i^\star| &\leq |\epsilon_i^{(t)}|+\sum_{j\in S,j\neq i}|W_{i,:}^{(t)}A_{:,j}(x_j^{(t)}+\epsilon_j^{(t)}-x_j^\star)|+\theta^{(t)} \\
&\leq |\epsilon_i^{(t)}|+\mu\sum_{j\in S,j\neq i}(|x_j^{(t)}-x_j^\star|+|\epsilon_j^{(t)}|)+\theta^{(t)}.
\end{aligned}
$$

Then, we have

$$
\begin{aligned}
\|x^{(t+1)}-x^\star\|_1 &\leq \sum_{i\in S}(|\epsilon_i^{(t)}|+\mu\sum_{j\in S,j\neq i}(|x_j^{(t)}-x_j^\star|+|\epsilon_j^{(t)}|)+\theta^{(t)}) \\
&= \|\epsilon^{(t)}\|_1+\mu(s-1)(\|x^{(t)}-x^\star\|_1+\|\epsilon^{(t)}\|_1)+s\theta^{(t)} \\
&= (1+\mu(s-1))\|\epsilon^{(t)}\|_1+\mu(s-1)\|x^{(t)}-x^\star\|_1+s\theta^{(t)}.
\end{aligned}
\tag{31}
$$

With equation 29, we have

$$
\|x^{(t+1)}-x^\star\|_1 \leq \rho^{(t)}C_2\|x^{(t)}-x^\star\|_1+\sum_{v=1}^{t-1}C_2^{t-v+1}\prod_{b=v+1}^{t}\rho^{(b)}\|x^{(v)}-x^\star\|_1+C_1\|x^{(t)}-x^\star\|_1+s\theta^{(t)},
\tag{32}
$$

where $C_1=\mu(s-1), C_2=1+C_1$.

**Step 3: Error Bound For The Whole Data Set.**
Finally, we take supremum over $x^\star \in \mathcal{X}(B, s)$,

$$\sup_{x^\star \in \mathcal{X}(B,s)} \{\|x^{(t+1)} - x^\star\|_1\} \leq \rho^{(t)} C_2 \sup_{x^\star \in \mathcal{X}(B,s)} \{\|x^{(t)} - x^\star\|_1\}$$

$$+ \sum_{v=1}^{t-1} C_2^{t-v+1} \prod_{b=v+1}^{t} \rho^{(b)} \sup_{x^\star \in \mathcal{X}(B,s)} \{\|x^{(v)} - x^\star\|_1\} \quad (33)$$

$$+ \sup_{x^\star \in \mathcal{X}(B,s)} \{C_1 \|x^{(t)} - x^\star\|_1\} + s\theta^{(t)}.$$

With equation A.1.1, we have

$$\sup_{x^\star \in \mathcal{X}(B,s)} \{\|x^{(t+1)} - x^\star\|_1\} \leq \rho^{(t)} C_2 \sup_{x^\star \in \mathcal{X}(B,s)} \{\|x^{(t)} - x^\star\|_1\}$$

$$+ \sum_{v=1}^{t-1} C_2^{t-v+1} \prod_{b=v+1}^{t} \rho^{(b)} \sup_{x^\star \in \mathcal{X}(B,s)} \{\|x^{(v)} - x^\star\|_1\}$$

$$+ C_1 \sup_{x^\star \in \mathcal{X}(B,s)} \{\|x^{(t)} - x^\star\|_1\} \quad (34)$$

$$+ (\mu + \mu\rho^{(t)})s \sup_{x^\star \in \mathcal{X}(B,s)} \{\|x^{(t)} - x^\star\|_1\}$$

$$+ \mu s \sum_{v=1}^{t-1} C_2^{t-v} \prod_{b=v+1}^{t} \rho^{(b)} \sup_{x^\star \in \mathcal{X}(B,s)} \{\|x^{(v)} - x^\star\|_1\}.$$

Since $\rho^{(t)}$ is a enough small scalar, we rearrange the above equation as follows:

$$\sup_{x^\star \in \mathcal{X}(B,s)} \{\|x^{(t+1)} - x^\star\|_1\} \leq H \sup_{x^\star \in \mathcal{X}(B,s)} \{\|x^{(t)} - x^\star\|_1\} + \mu \sum_{v=1}^{t-1} C_3^{t-v} \sup_{x^\star \in \mathcal{X}(B,s)} \{\|x^{(v)} - x^\star\|_1\},$$

$$(35)$$

where $H = (2s - 1)\mu(1 + \bar{\rho}) + \bar{\rho}$, $C_3 = \bar{\rho}(1 + C_2)$, and $\bar{\rho}$ is the upper bound of $\rho^{(t)}$ for all $t$. By induction, with $c = -\log(H)$, we have

$$\sup_{x^\star \in \mathcal{X}(B,s)} \{\|x^{(t+1)} - x^\star\|_1\} \leq (H^t + r(t, \mu, s, \rho)) \sup_{x^\star \in \mathcal{X}(B,s)} \{\|x^{(t)} - x^\star\|_1\}$$

$$\leq \sup_{x^\star \in \mathcal{X}(B,s)} \{\|x^{(0)} - x^\star\|_1\} \leq sB(\exp(-ct) + r(t, \mu, s, \rho)),$$

where $r(t, \mu, s, \rho)$ donates the slight influence of the second term of equation 35. Since $\|x\|_2 \leq \|x\|_1, \forall x \in \mathbb{R}$, we can get the upper bound for $l_2$ norm:

$$\sup_{x^\star \in \mathcal{X}(B,s)} \{\|x^{(t+1)} - x^\star\|_2\} \leq \sup_{x^\star \in \mathcal{X}(B,s)} \{\|x^{(t+1)} - x^\star\|_1\} \leq sB(\exp(-ct) + r(t, \mu, s, \rho)),$$

As long as $s \leq ((1 - \rho)/((1 + \rho)\mu) + 1)/2$, $c = -log(H) > 0$, then the error bound holds uniformly for all $x^\star \in \mathcal{X}(B, s)$.

### A.1.2  PROOF FOR SALISTA WITH $\beta = 1$

When $\beta \neq 0$,

$$\frac{\rho^{(t)}}{\|g^{(t)}\|_2}$$

is not small enough, which means something new is needed. Therefore, we need to further explore $\|\epsilon^{(t)}\|_1$:

$$\|\epsilon^{(t)}\|_2 = \frac{\rho^{(t)}}{\|m^{(t)} \odot g^{(t)}\|_2} \|\|m^{(t)} \odot g^{(t)}\|_2 = \rho^{(t)} \geq \frac{1}{\sqrt{s}} \|\epsilon^{(t)}\|_1. \quad (36)$$

**Step 1:** *No False Positives.*
Let $S = \text{support}(x^\star)$ indicates the non-zero entires. We want to prove by induction that, as long as all trained

$W^{(t)}$ satisfies the "good" conditions in Definition 4, $x_i^{(t)} = 0, i \notin S$ (*no false positives*). As we set $x^{(0)} = 0$, it is satisfied when $t = 0$ and

$$\theta^{(t)} = \mu \sup_{x^\star \in \mathcal{X}(B,s)} \{\|x^{(t)} - x^\star\|_1\} + \sqrt{s}\mu\rho^{(t)}, \tag{37}$$

where $\mathcal{X}(B,s) = \{x^\star \mid |x_i^\star| \le B, \forall i, \|x\|_0 \le s\}$ is defined in the Basic Assumption 1. Fixing $t$ and assuming $x_i^{(v)} = 0, i \notin S, \forall v \in \mathbb{N}^+ \le t$, then we have

$$x_i^{(t+1)} = \eta_{\theta^{(t)}}(x_i^{(t)} + \epsilon_i^{(t)} - W_{i,:}^{(t)}(A(x^{(t)} + \epsilon^{(t)}) - y)) - \epsilon_i^{(t)}$$
$$= \eta_{\theta^{(t)}}(-W_{i,:}^{(t)}(A(x^{(t)} + \epsilon^{(t)}) - y)), i \notin S,$$

where $\epsilon_i^{(t)} = 0$ as the mask in (27). Since and $W^{(t)}$ is good, we have

$$\begin{aligned}
\theta^{(t)} &\ge \mu\|x^{(t)} - x^\star\|_1 + \sqrt{s}\mu\rho^{(t)} \\
&\ge \mu(\|x^{(t)} - x^\star\|_1 + \|\epsilon_j^{(t)}\|_1) \\
&\ge \sum_{j \in S}(|W_{i,:}^{(t)}A_{:j}(x_i^{(t)} - x_j^\star)| + |W_{i,:}^{(t)}A_{:j}\epsilon_j^{(t)}|) \\
&\ge \sum_{j \in S}|W_{i,:}^{(t)}A_{:j}(x_j^{(t)} + \epsilon_j^{(t)} - x_j^\star)|, \forall i \in S,
\end{aligned} \tag{38}$$

where we can achieve (38) with Eq. (36).

**Step 2: Upper Bound of Recovery Error.**

Since Eq. (36) has no impact on the update rule (24), we can follow the conclusion of (31), and thus we have

$$\begin{aligned}
\|x^{(t+1)} - x^\star\|_1 &\le (1 + \mu(s-1))\|\epsilon^{(t)}\|_1 + \mu(s-1)\|x^{(t)} - x^\star\|_1 + s\theta^{(t)} \\
&\le (1 + \mu(s-1))\sqrt{s}\rho^{(t)} + \mu(s-1)\|x^{(t)} - x^\star\|_1 + s\theta^{(t)}.
\end{aligned} \tag{39}$$

**Step 3: Error Bound For The Whole Data Set.**
Finally, we take supremum over $x^\star \in \mathcal{X}(B,s)$,

$$\sup_{x^\star \in \mathcal{X}(B,s)} \{\|x^{(t+1)} - x^\star\|_1\} \le \mu(s-1) \sup_{x^\star \in \mathcal{X}(B,s)} \{\|x^{(t)} - x^\star\|_1\} + (1 + \mu(s-1))\sqrt{s}\rho^{(t)} + s\theta^{(t)}. \tag{40}$$

With equation 37, we have

$$\sup_{x^\star \in \mathcal{X}(B,s)} \{\|x^{(t+1)} - x^\star\|_1\} \le \rho^{(t)}\mu(2s-1) \sup_{x^\star \in \mathcal{X}(B,s)} \{\|x^{(t)} - x^\star\|_1\} + (1 + \mu(2s-1))\sqrt{s}\rho^{(t)}. \tag{41}$$

In this case, we have $H = (2s-1)\mu, C = (1 + \mu(2s-1))\sqrt{s}\bar{\rho}$, and $\bar{\rho}$ is the upper bound of $\rho^{(t)}$ for all $t$. By induction, with $c = -\log(H)$, we have

$$\begin{aligned}
\sup_{x^\star \in \mathcal{X}(B,s)} \{\|x^{(t+1)} - x^\star\|_1\} &\le H \sup_{x^\star \in \mathcal{X}(B,s)} \{\|x^{(t+1)} - x^\star\|_1\} + C \\
&\le sB\exp(-ct) + C\sum_{\tau=0}^{k+1}(H^\tau) \\
&\le sB\exp(-ct) + \frac{C}{1-H}.
\end{aligned}$$

Since $\|x\|_2 \le \|x\|_1, \forall x \in \mathbb{R}$, we can get the upper bound for $l_2$ norm:

$$\sup_{x^\star \in \mathcal{X}(B,s)} \{\|x^{(t+1)} - x^\star\|_2\} \le \sup_{x^\star \in \mathcal{X}(B,s)} \{\|x^{(t+1)} - x^\star\|_1\} \le sB\exp(-ct) + \frac{1+H}{1-H}\sqrt{s}\bar{\rho},$$

As long as $s \le (1/\mu + 1)/2, c = -\log(H) > 0$, then the error bound holds uniformly for all $x^\star \in \mathcal{X}(B,s)$.

### A.1.3   PROOF FOR SALISTA WITH $\beta \in (0,1)$

When $\beta \in (0,1)$, each update of $u$ can be devide to $\beta = 1$ and $\beta = 0$. Therefore, its convergence property lies between the two cases mentioned above. $\qquad\square$

## A.2 Additional Experiment and Details

In some fields, such as synthetic aperture processing (Li et al., 2025), the DUNs paradigm based on ISTA-NET (Zhang & Ghanem, 2018) still attracts considerable attention. Therefore, we apply the SADUN framework proposed in this paper to ISTA-NET to ensure the universality of our framework. For the convenience of characterizing the model, we denote a single convolution operator as $c(x)$ and some composite operations $cr(x) = relu(c(x)), cbr(x) = relu(bn(c(x)))$.

### A.2.1 Details of Sparse Coding Task

In sparse coding task, we choose $m = 250, n = 500$. We sample the entries of $A$ i.i.d. from the standard Gaussian distribution, $Aij \sim N(0, 1/m)$ and then normalize its columns to have the unit $l_2$ norm. We fix a matrix A in each setting where different networks are compared. To generate sparse vectors $x^\star$, we decide each of its entry to be non-zero following the Bernoulli distribution with pb = 0.1. The values of the non-zero entries are sampled from the standard Gaussian distribution. A test set of 1000 samples generated in the above manner is fixed for all tests in our simulations. And we use multi-stage training strategy (Chen et al., 2018; Liu et al., 2018) to train our SALISTA-CP, SALISTA-CPSS and SALISTA-ANA. Moreover, LISTA-CPSS, LISTA-ANA and our SADUNs version all use the support selection technique:

$$\eta_{\theta^{(t)}}^{p^{(t)}}(x_i) = \begin{cases} x_i, & i \in S^{p^{(k)}}(x) \\ 0, & |x_i| \leq \theta^{(t)} \\ \eta_{\theta^{(t)}}(x_i), & otherwise, \end{cases} \tag{42}$$

where $S^{p^{(k)}}(x)$ includes the elements with the largest pk% magnitudes in vector $x$.

### A.2.2 Details of SISR Task

For the SISR problem (22) and other model contains conventional operator, the fourier transform is usually used in its closed-form solution. For model (22), the closed-form solution is defined as:

$$\mathcal{F}^{-1}\left(\frac{1}{\alpha^{(t)}}\left(d - \overline{\mathcal{F}(t)} \odot_s \frac{\mathcal{F}(t)d \Downarrow_s}{\overline{\mathcal{F}(t)}\mathcal{F}(t) \Downarrow_s +\alpha^{(t)}}\right)\right),$$

where $d = \overline{\mathcal{F}(t)}\mathcal{F}(y \Uparrow_s) + \alpha^{(t)}\mathcal{F}(z^{(t)})$ and $\odot_s$ denotes distinct block processing operator with element-wise multiplication, $\Downarrow_s$ denotes distinct block downsampler, $\overline{x}$ means the conjugate transpose of $x$. And the architecture of IRCNN is defined as

$$prox_{\lambda/\mu g}(x) = x + cbr(cbr(cbr(cbr(cr(x))))).$$

### A.2.3 Full Visual Comparisons of the experiments based on ISTA-NET

Due to page limitations, we have only presented a portion of the visual comparison results in the main text. In Figure 4, we present all the results for better comparison.

## A.3 Discussion on the Proposed Modules

For deep unfolding networks, the data fidelity term $f$ is often determined by downstream tasks. Therefore, more emphasis is placed on designing a well-performing regularization term $g$, or rather, the proximal operator of the regularization term. In our SADUNs framework, these two components are defined as DFM and PMM respectively, to facilitate their application in different scenarios.

For simple optimization problems, such as sparse coding, it is actually unnecessary to use (16) for approximation; instead, the subgradient can be used directly for calculatio by

$$g^{(t)} = \nabla f(x^{(t)}) + \lambda sign(x^{(t)}) \tag{43}$$

where $sign(x^{(t)}) \in \partial g(x^{(t)})$. We compare this strategy of directly using SAM with our proposed scheme, as shown in Figure 5. We also compared the number of parameters and running speed between SADUNs and SAM+LISTA, as shown in Table 6.However, for complex application problems,

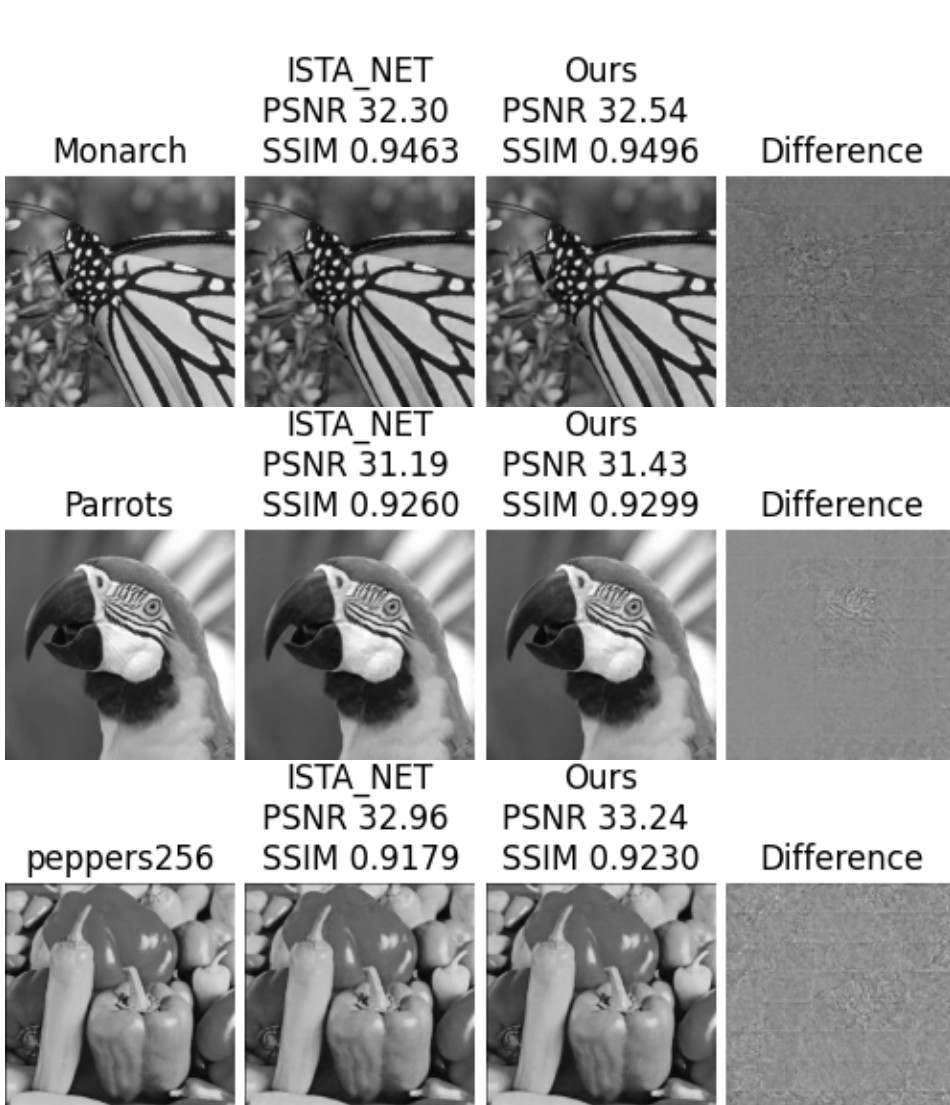

Figure 4: Visual Comparisons of the experiments based on ISTA-NET

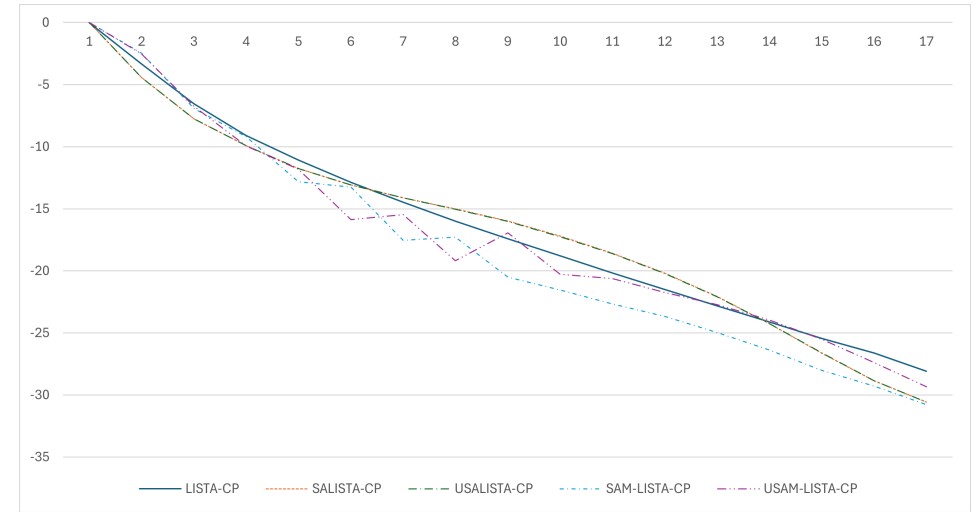

Figure 5: Comparisons of sparse representation between our approximation and real subgradient

the subgradient of the regularization term is difficult to calculate directly, which means that the improvement of SCM is relatively challenging. Finally, in this paper, the SPM adopts the update form of Unified SAM to achieve a balance between SAM and USAM. In addition, other SAM variants can also be adopted, such as ASAM. We further attempted to use the update form of ASAM in SPM:

$$\epsilon^{(t)} = \rho^{(t)} \frac{(x^{(t+1)})^2 * g^{(t+1)}}{\|x^{(t+1)} * g^{(t+1)}\|_2},$$

and conducted a brief comparison as shown in Figure 6.

Table 6: Comparison of the network structures and running speed between SADUNs and SAM+LISTA.

|  | SAM-LISTA-CP | USAM-LISTA-CP | SALISTA-CP | USALISTA-CP |
|---|---|---|---|---|
| number of parameters | 2MN+3 | 2MN+3 | MN+3 | MN+3 |
| running speed (s) | 2.56 | 2.57 | 1.83 | 1.85 |

## A.4 Details the Usage of Large Language Models

We conducted a simple review and grammar check by LLM model.

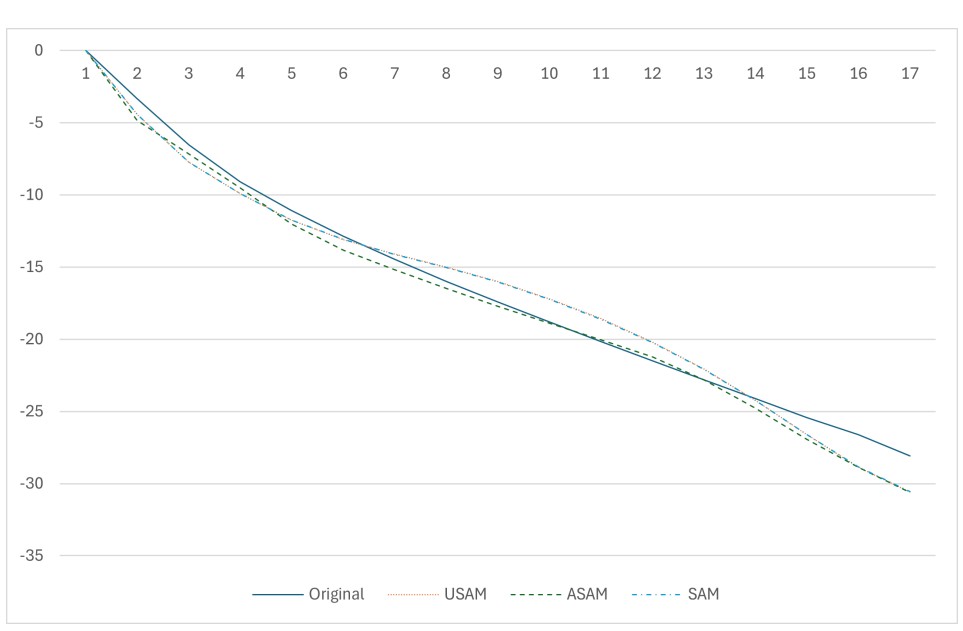

Figure 6: Comparisons of sparse representation between Unified SAM and ASAM

