# OpenReview forum: "SADUNs: Sharpness-Aware Deep Unfolding Networks for Image Restoration"
_ICLR.cc/2026/Conference — Submitted to ICLR 2026_

### Official Review · Reviewer_MvHm · 2025-10-31

**Soundness:** 2
**Presentation:** 3
**Contribution:** 1
**Rating:** 2
**Confidence:** 4

**Summary:**

This paper investigates sharpness-aware minimization (SAM) in the context of deep unfolding networks (DUN) for inverse problems. They start from a minimization problem that consists of two terms: data fidelity and the SAM cost, which serves as a regularization to promote minima in flatter regions of the landscape, aiming to achieve better generalization. They propose a solver for the problem based on the majorization-minimization technique. They then design an unfolded architecture based on this solver. The idea is nice, but it is very incremental, and in my opinion, the authors fail to demonstrate the advantage of the SAM within the unfolding scheme, as the performance gains are very marginal. Additionally, they do not present any qualitative results, e.g., to facilitate visual comparisons, which is essential, in my opinion, for a paper on an image reconstruction topic. Overall, the paper provides an interesting perspective but does not convincingly demonstrate the practical advantages of incorporating SAM into DUNs.

**Strengths:**

The paper is well written, easy to follow, and the method seems to be correct theoretically, and a nice direction to explore.

**Weaknesses:**

The contribution is incremental, with no significant improvement in performance or efficiency over existing deep unfolding networks. The paper also lacks visual comparisons. Overall, the experimental evidence is not convincing that incorporating SAM provides a meaningful advantage within the unfolding framework.

**Questions:**

While a method does not need large numerical gains to be valuable, such claims should be precisely stated and justified. The first line of the conclusion mentions “significant performance improvements,” but the tables show only marginal gains. Could the authors clarify what specific improvements they consider significant? Are they referring to convergence behavior, stability, or something beyond PSNR/SSIM? In what scenarios would adding SAM be preferable? For example, again, in terms of robustness, generalization to unseen data, could you design an experiment to show this? If yes, it would be beneficial to include such experiments in the main body of text to show those advantages.

---

> ### Author Response · Authors · 2025-11-23
> **Rebuttal by Authors**
>
> Dear Reviewer MvHm
>
>
> We sincerely thank you for your detailed comments and thoughtful and constructive feedback. We hope our response adequately addresses your concerns and restores your confidence in our work. We look forward to hearing your thoughts.
>
>
> For W1 & Q2, in the research on image spatial filtering, image smoothing and sharpening are classic topics. In [TV-layer], the authors designed a learnable proximal operator based on TV regularization and proposed a scheme for adjusting image sharpness according to the properties of TV regularization. However, the proximal operators used in existing DUNs are often black-box models (e.g., U-shaped networks [LTWIST], various attention mechanisms [UFC-NET], and additional residual connections [UFC-NET, Heros]). This means that we can only understand these proximal operators (network modules) in an abstract way (for instance, [RED] interprets them as denoisers). Moreover, the input of these models usually only contains zero-order information (images) and does not include differential information or the like, which implies that the sharpness perception of these modules is relatively weak. We aim to introduce the SAM algorithm to provide a new and extensible scheme for DUNs to extract image texture features.
>
>
> We have added this part of the motivation and the accompanying experiments in the main text, including:
> 1.	Whether the performance of the expanded network's output can be improved through smoothing and sharpening.
> 2.	The difference between the outputs of SADUNs and DUNs, to ensure that we have indeed obtained more edge information.
>
>
> For Q1, our framework demonstrates almost consistent performance improvement over ISTA-NET, which only uses convolutional layers, and UFC-NET, which incorporates advanced mechanisms such as CNV2 and multi-head attention. This proves that existing DUNs have weak ability to extract sharpness or curvature information, and proposes a feasible and universal solution.
>
>
> Although in the field of DUNs, the Peak Signal-to-Noise Ratio (PSNR) reported in top-tier conferences can often be improved by 0.2-0.4, these models are usually standalone, and their improvements stem from the development of deep learning technologies rather than the exploration of flaws in
> DUNs. Therefore, we place greater emphasis on demonstrating the consistency of our algorithm, so as to ensure that exploring the sharpness information of images is a feasible solution for the subsequent development of DUNs.
>
>
> For Q3, in terms of data, we tend to use data with relatively complex texture features. This is because the sharpness-aware module we designed is mainly used to enhance edge information. The fact that UFC-NET shows a relatively small performance improvement on the Urban100 dataset indirectly confirms this point.
> Regarding application scenarios, this improvement should be generally effective for current DUNs based on ISTA or HQS. Before better learnable TV regularization or similar technologies are proposed, using sharpness-aware minimization to indirectly improve image sharpness is a relatively feasible solution at present.
>
>
> We have revised the content of the motivation, background, and experimental sections of the paper, and expanded the main text to 10 pages. Thank you once again for reviewing our work. We hope that our responses and clarifications will facilitate a more favorable evaluation of our paper.
>
> Reference:
> [TV-layer] YEH RaymondA, HU Y T, REN Z, et al. Total Variation Optimization Layers for Computer Vision[J]. 2022.
>
> [LTWIST] Gan H, Wang X, He L, et al. Learned two-step iterative shrinkage thresholding algorithm for deep compressive sensing[J]. IEEE Transactions on Circuits and Systems for Video Technology, 2023, 34(5): 3943-3956.
>
> [UFC-NET] WANG X, GAN H. UFC-Net: Unrolling Fixed-point Continuous Network for Deep Compressive Sensing[J].
> [Heros] ZHANG X, ZHANG Y, XIONG R, et al. HerosNet: Hyperspectral Explicable Reconstruction and Optimal Sampling Deep Network for Snapshot Compressive Imaging[J].
>
> [RED] ROMANO Y, ELAD M, MILANFAR P. The Little Engine That Could: Regularization by Denoising (RED)[J/OL]. SIAM Journal on Imaging Sciences, 2017: 1804-1844. http://dx.doi.org/10.1137/16m1102884. DOI:10.1137/16m1102884.

---

### Official Review · Reviewer_WntY · 2025-10-31

**Soundness:** 3
**Presentation:** 2
**Contribution:** 3
**Rating:** 4
**Confidence:** 3

**Summary:**

This paper proposes Sharpness-Aware Deep Unfolding Networks (SADUNs) — a new framework that enhances traditional Deep Unfolding Networks (DUNs) by incorporating Sharpness-Aware Minimization (SAM) into the proximal operator optimization process.

**Strengths:**

Standard DUNs (like LISTA, ISTA-Net, UFC-Net) suffer from optimization instability and limited adaptability when integrating complex black-box priors. The paper introduces sharpness awareness into DUNs to improve convergence and generalization. The framework redefines each DUN iteration as a sharpness-perturbed proximal step, replacing redundant gradient computations via proximal operator properties. The authors show that, under sparse coding assumptions, SADUNs achieve linear convergence

**Weaknesses:**

Most experiments are still within linear inverse problems (CS, SISR). There is no evaluation on nonlinear or dynamic inverse tasks (e.g., MRI, deblurring, or radar), where SAM’s stability might differ.

Gains over strong baselines (e.g., UFC-Net) are minor (≈ 0.1 dB PSNR, < 0.002 SSIM), raising questions about practical significance despite theoretical novelty.

The authors claim no inference slowdown, but runtime and FLOPs comparisons are absent.
Quantitative verification (speed vs accuracy trade-off) would strengthen the paper’s engineering relevance.

**Questions:**

1. How sensitive is SADUN performance to the choice of subgradient approximation (Property 2)? Could replacing ∇̃g with backprop-based gradients further improve stability?

2. The authors are encouraged to empirically verify whether the “flatter minima” achieved via SAM actually translate to better generalization on unseen degradations.

3. The authors are encouraged to include visual comparison results to better illustrate the qualitative advantages of SADUNs over baseline methods.

4. The ablation analysis for SAM hyperparameters (ρ, β) is minimal; only a few static values are shown in tables. A more systematic study (varying ρ/β → performance → stability) would clarify the effect of sharpness control.

---

> ### Author Response · Authors · 2025-11-23
> **Rebuttal by Authors**
>
> Dear Reviewer WntY
>
> We sincerely thank you for your detailed comments and thoughtful and constructive feedback. We hope our response adequately addresses your concerns and restores your confidence in our work. We look forward to hearing your thoughts.
>
>
> For W1, most experiments are still within linear inverse problems (CS, SISR).
>
>
> Yes, this is because most DUNs are modeled based on linear inverse problems. For this paper, we mainly focuses on ISTA-based DUNs. This implies that nonlinear inverse problems do not necessarily meet the requirements of ISTA and our approximation. Furthermore, the adoption of nonlinear approximation in models such as HerosNet is often due to excessively low sampling rates, and their analysis is still based on linear models.
>
>
> For W2 & W3, gains over strong baselines (e.g., UFC-Net) are minor.
>
>
> Although in the field of DUNs, the Peak Signal-to-Noise Ratio (PSNR) reported in top-tier conferences can often be improved by 0.2-0.4, these models are usually standalone, and their improvements stem from the development of deep learning technologies rather than the exploration of flaws in DUNs. Therefore, we place greater emphasis on demonstrating the consistency of our algorithm, so as to ensure that exploring the sharpness information of images is a feasible solution for the subsequent development of DUNs.
>
>
> The framework we propose does not involve the core deep neural network modules. Moreover, we only add two parameters per layer—this number is far smaller than the number of parameters in the proximal operator within each layer. Taking the simplest DUNs (ISTA-NET) as an example, its parameters per layer include the learning rate, threshold, and four convolutional kernels, the total parameters number is
>
> 1+1+3\*2\*1\*3\*3+32\*32\*3\*3\*2+1\*32\*3\*3=19010.
>
> Therefore, the increase in the model's parameter count is negligible (2/19010).
>
>
> Due to the large number of custom operators contained in the expanded network, calculating the FLOPs (Floating-Point Operations Per Second) becomes relatively difficult. We conducted 1000 tests and provided the average running time on CPU and GPU. The results are as follows: ISTA-NET: 0.024/0.0036, SAISTA-NET: 0.025/0.0059.
>
>
> For Q1: the DUNs are defined as bi-level optimization problems. Since the regularizer in the inner-level optimization problem is abstract, there is almost no way to use it as a loss for backpropagation, which in turn renders such an approximation necessary. Currently, another feasible solution is to adopt INN (Implicit Neural Network) as the proximal operator, though this approach is not yet prevalent.
>
>
> For Q2 & Q3: in the research on image spatial filtering, image smoothing and sharpening are classic topics. In [TV-layer], the authors designed a learnable proximal operator based on TV regularization and proposed a scheme for adjusting image sharpness according to the properties of TV regularization. However, the proximal operators used in existing DUNs are often black-box models (e.g., U-shaped networks [LTWIST], various attention mechanisms [UFC-NET], and additional residual connections [UFC-NET, Heros]). This means that we can only understand these proximal operators (network modules) in an abstract way (for instance, [RED] interprets them as denoisers). Moreover, the input of these models usually only contains zero-order information (images) and does not include differential information or the like, which implies that the sharpness perception of these modules is relatively weak. We aim to introduce the SAM algorithm to provide a new and extensible scheme for DUNs to extract image texture features.
>
>
>
> We have revised the content of the motivation, background, and experimental sections of the paper, and expanded the main text to 10 pages. Thank you once again for reviewing our work. We hope that our responses and clarifications will facilitate a more favorable evaluation of our paper.
>
>
> Reference:
> [TV-layer] YEH RaymondA, HU Y T, REN Z, et al. Total Variation Optimization Layers for Computer Vision[J]. 2022.
>
> [LTWIST] Gan H, Wang X, He L, et al. Learned two-step iterative shrinkage thresholding algorithm for deep compressive sensing[J]. IEEE Transactions on Circuits and Systems for Video Technology, 2023, 34(5): 3943-3956.
>
> [UFC-NET] WANG X, GAN H. UFC-Net: Unrolling Fixed-point Continuous Network for Deep Compressive Sensing[J].
>
> [Heros] ZHANG X, ZHANG Y, XIONG R, et al. HerosNet: Hyperspectral Explicable Reconstruction and Optimal Sampling Deep Network for Snapshot Compressive Imaging[J].
>
> [RED] ROMANO Y, ELAD M, MILANFAR P. The Little Engine That Could: Regularization by Denoising (RED)[J/OL]. SIAM Journal on Imaging Sciences, 2017: 1804-1844. http://dx.doi.org/10.1137/16m1102884. DOI:10.1137/16m1102884.

---

### Official Review · Reviewer_Wr7b · 2025-11-03

**Soundness:** 2
**Presentation:** 2
**Contribution:** 2
**Rating:** 4
**Confidence:** 3

**Summary:**

This paper proposes SADUNs, a framework that integrates Sharpness-Aware Minimization (SAM) into Deep Unfolding Networks (DUNs) for image restoration tasks. The key idea is to use the properties of proximal operators to efficiently compute a subgradient approximation, which is then used within a Unified SAM formulation to perturb the optimization landscape. This design aims to improve performance without increasing inference time and allows for fine-tuning from pre-trained DUNs. The authors provide a theoretical analysis showing linear convergence for a sparse coding variant (SALISTA-CP) and demonstrate empirical improvements on tasks including synthetic sparse coding, natural image compressive sensing, and single image super-resolution.

**Strengths:**

A significant advantage of the proposed method is its claim of not degrading inference speed, which is achieved by reusing computations from the unfolding step. The fine-tuning compatibility is also a practical benefit for adapting existing models.

The paper provides extensive experiments across multiple tasks (sparse coding, CS, SISR) and several base DUN architectures (LISTA variants, UFC-NET, ISTA-NET), demonstrating the general applicability of the framework.

**Weaknesses:**

The actual performance gains reported are often marginal. In many cases (e.g., Table 1, Table 3, Table 4), the improvements in PSNR/SSIM are fractions of a decibel or tiny absolute increments. While consistent, these gains are arguably too small to constitute a major advance. The paper lacks a compelling demonstration of a scenario where SADUNs provide a substantial qualitative or quantitative leap over strong baselines, which is critical for a top-tier conference publication.

The paper successfully demonstrates the "what" (improvements are possible) but falls short on the "why." There is a lack of analysis connecting the sharpness-aware formulation to the observed improvements in image restoration quality. How does the perturbation induced by SADUNs specifically improve the texture generation, artifact suppression, or detail recovery in the output images? A qualitative analysis comparing the loss landscapes or the behavior of attention maps (in transformer-based DUNs) with and without SADUNs would significantly strengthen the claims.

**Questions:**

No Questions

---

> ### Author Response · Authors · 2025-11-23
> **Rebuttal by Authors**
>
> Dear Reviewer Wr7b
>
> We sincerely thank you for your detailed comments and thoughtful and constructive feedback. We hope our response adequately addresses your concerns and restores your confidence in our work. We look forward to hearing your thoughts.
>
>
> For W1, our paper successfully demonstrates the "what" (improvements are possible) but falls short on the "why."
>
>
> In the research on image spatial filtering, image smoothing and sharpening are classic topics. In [TV-layer], the authors designed a learnable proximal operator based on TV regularization and proposed a scheme for adjusting image sharpness according to the properties of TV regularization. However, the proximal operators used in existing DUNs are often black-box models (e.g., U-shaped networks [LTWIST], various attention mechanisms [UFC-NET], and additional residual connections [UFC-NET, Heros]). This means that we can only understand these proximal operators (network modules) in an abstract way (for instance, [RED] interprets them as denoisers). Moreover, the input of these models usually only contains zero-order information (images) and does not include differential information or the like, which implies that the sharpness perception of these modules is relatively weak. We aim to introduce the SAM algorithm to provide a new and extensible scheme for DUNs to extract image texture features.
>
>
> For W2, how does the perturbation induced by SADUNs specifically improve the texture generation, artifact suppression, or detail recovery in the output images?
>
>
> The introduction of our SAM is intended to improve certain edge features, thereby enhancing the consistency between the output and the original image. In the field of adversarial examples, when deep neural networks perform classification tasks, even minor changes in edge features may affect the networks' classification results. Through sharpness-aware minimization technology, our framework can better preserve such information to facilitate its use in downstream tasks.
>
>
> For W3. the actual performance gains reported are often marginal.
>
>
> Due to space limitations, to demonstrate the application and extensibility of our model in the new algorithm, we have listed the experiments based on ISTA-NET in the appendix. Our scheme can improve the PSNR of ISTA-NET and UFC-NET by 0.1 on set11 in compressed sensing experiments, with the only cost being the addition of a few addition and subtraction operations and a parameter count far smaller than that of the original model. The experiment on UFC-NET is intended to demonstrate our framework is effective on state-of-the-art models. Even when advanced modules such as CNV2 and multi-head attention are adopted, DUNs can still improve their performance by enhancing sharpness—this effectiveness is not limited to simple convolutional neural networks (CNNs) alone. Although in the field of DUNs, the Peak Signal-to-Noise Ratio (PSNR) reported in top-tier conferences can often be improved by 0.2-0.4, these models are usually standalone, and their improvements stem from the development of deep learning technologies rather than the exploration of flaws in DUNs. Therefore, we place greater emphasis on demonstrating the consistency of our algorithm, so as to ensure that exploring the sharpness information of images is a feasible solution for the subsequent development of DUNs.
>
>
>
> We have revised the content of the motivation, background, and experimental sections of the paper, and expanded the main text to 10 pages. Thank you once again for reviewing our work. We hope that our responses and clarifications will facilitate a more favorable evaluation of our paper.
>
>
> Reference:
> [TV-layer] YEH RaymondA, HU Y T, REN Z, et al. Total Variation Optimization Layers for Computer Vision[J]. 2022.
>
> [LTWIST] Gan H, Wang X, He L, et al. Learned two-step iterative shrinkage thresholding algorithm for deep compressive sensing[J]. IEEE Transactions on Circuits and Systems for Video Technology, 2023, 34(5): 3943-3956.
>
> [UFC-NET] WANG X, GAN H. UFC-Net: Unrolling Fixed-point Continuous Network for Deep Compressive Sensing[J].
>
> [Heros] ZHANG X, ZHANG Y, XIONG R, et al. HerosNet: Hyperspectral Explicable Reconstruction and Optimal Sampling Deep Network for Snapshot Compressive Imaging[J].
> [RED] ROMANO Y, ELAD M, MILANFAR P. The Little Engine That Could: Regularization by Denoising (RED)[J/OL]. SIAM Journal on Imaging Sciences, 2017: 1804-1844. http://dx.doi.org/10.1137/16m1102884. DOI:10.1137/16m1102884.

---

### Author Response · Authors · 2025-11-29
**Summary of Rebuttal**

Dear New Area Chair,

We sincerely appreciate the additional time and effort you have dedicated to reviewing our work.

The comments from the three reviewers are relatively consistent, mainly covering the following aspects:

1.	The **reasons** for the performance improvement achieved by introducing SAM are neither described.

    In the research on image spatial filtering, image smoothing and sharpening are classic topics. In [TV-layer], the authors designed a learnable proximal operator based on TV regularization and proposed a scheme for adjusting image sharpness according to the properties of TV regularization. However, the proximal operators used in existing DUNs are often black-box models (e.g., U-shaped networks [LTWIST], various attention mechanisms [UFC-NET], and additional residual connections [UFC-NET, Heros]). This means that we can only understand these proximal operators (network modules) in an abstract way (for instance, [RED] interprets them as denoisers). Moreover, the input of these models usually only contains zero-order information (images) and does not include differential information or the like, which implies that the sharpness perception of these modules is relatively weak. We aim to introduce the SAM algorithm to provide a new and extensible scheme for DUNs to extract image texture features.

    We have supplemented **the visual comparison results** in the appendix, with a brief presentation provided in the main text.

2.	**The improvement in PSNR (Peak Signal-to-Noise Ratio) is limited**.

    We propose a **framework-based algorithm** that leverages insights from the Iterative Shrinkage-Thresholding Algorithm (ISTA). Specifically, we ingeniously integrate Sharpness-Aware Minimization (SAM) into the ISTA framework, which is then unfolded into a deep unfolding network (DUN) comprising four modules. Our framework, termed Sharpness-Aware Deep Unfolding Networks (SADUNs), enhances the utilization of sharpness information in DUNs with only a marginal increase in parameters and computational complexity. Notably, it achieves universal performance improvements across DUN variants employing traditional proximal operators (e.g., LISTA), simple convolutional operators (e.g., ISTA-NET), multi-head attention mechanisms, and advanced network architectures (e.g., UFC-NET).

    Most existing related works achieve performance gains primarily by designing novel network modules or adopting stronger baseline algorithms, without addressing the limitations inherent in such black-box regularizers or proximal operators. Through experiments, we find that applying small sharpening or smoothing perturbations directly to the outputs of networks such as ISTA-NET, UFC-NET, and LTWIST does not necessarily lead to performance degradation. We have supplemented the results of perturbation experiments on ISTA-NET in Section 2.5.

3.	There is a lack of comparison regarding the number of parameters, runtime, and FLOPs (Floating-Point Operations per Second).

    The framework we propose does not involve the core deep neural network modules. Moreover, we only add two parameters per layer—this number is far smaller than the number of parameters in the proximal operator within each layer. Taking ISTA-NET as an example, its parameters per layer include the learning rate, threshold, and four convolutional kernels, the total parameters number is 19010. Therefore, the increase in the model's parameter count is negligible (2/19010). And we conducted 1000 tests and provided the average running time on CPU and GPU. The results are as follows: ISTA-NET: 0.024/0.0036, SAISTA-NET: 0.025/0.0059 which is update to Table 2.

Reference:

[TV-layer] YEH RaymondA, HU Y T, REN Z, et al. Total Variation Optimization Layers for Computer Vision[J]. 2022.

[LTWIST] Gan H, Wang X, He L, et al. Learned two-step iterative shrinkage thresholding algorithm for deep compressive sensing[J]. IEEE Transactions on Circuits and Systems for Video Technology, 2023, 34(5): 3943-3956.

[UFC-NET] WANG X, GAN H. UFC-Net: Unrolling Fixed-point Continuous Network for Deep Compressive Sensing[J].

[Heros] ZHANG X, ZHANG Y, XIONG R, et al. HerosNet: Hyperspectral Explicable Reconstruction and Optimal Sampling Deep Network for Snapshot Compressive Imaging[J]. [RED] ROMANO Y, ELAD M, MILANFAR P. The Little Engine That Could: Regularization by Denoising (RED)[J/OL]. SIAM Journal on Imaging Sciences, 2017: 1804-1844. http://dx.doi.org/10.1137/16m1102884. DOI:10.1137/16m1102884.

---

> ### Author Response · Authors · 2025-11-29
> **Summary of non-common issues**
>
> In addition, Reviewer 2 raised the following question: *"Could replacing ∇̃g with backprop-based gradients further improve stability?"*
>
>     The DUNs are defined as bi-level optimization problems. Since the regularizer in the inner-level optimization problem is abstract, there is almost no way to use it as a loss for backpropagation, which in turn renders such an approximation necessary. Currently, another feasible solution is to adopt INN (Implicit Neural Network) as the proximal operator, though this approach is not yet prevalent.
>
> Reviewer 3 also inquired: *"In what scenarios would adding SAM be preferable?"*
>
>     Considering only the data, we tend to introduce our SADUNs framework for scenarios with complex texture features. In the experiments on UFC-NET, the improvement on the Urban100 dataset is relatively minor, and this is because most of the images it contains are of glass curtain walls, whose textures are relatively simple. This also indirectly confirms that our SADUNs enhances the performance of existing unfolding networks by leveraging sharpness information.
>
>     Regarding application scenarios, this improvement should be generally effective for current DUNs based on ISTA or HQS. Before better learnable TV regularization or similar technologies are proposed, using sharpness-aware minimization to indirectly improve image sharpness is a relatively feasible solution at present.

---

### Meta-Review · Area_Chair_5rQF · 2025-12-25

**Summary:**

The paper proposes SADUNs, integrating Sharpness-Aware Minimization (SAM) into deep unfolding networks (DUNs) via proximal-operator properties, aiming to improve restoration quality while preserving inference speed and enabling fine-tuning from pre-trained DUNs. Reviewers agree the paper is generally readable and the idea is reasonable, but they raised substantial concerns about the practical significance and evidential support: the reported gains are often marginal (small PSNR/SSIM improvements), the paper does not sufficiently explain *why* SAM improves restoration in the unfolding setting, and the experimental validation is limited mainly to linear inverse problems. A reviewer also emphasized the incremental nature of the contribution and the lack of convincing qualitative evidence in the original submission. Unfortunately, I recommend **rejecting** this submission.

**Reviewer Concerns:**

**Concerns addressed by the rebuttal / discussion:**
- **Missing engineering evidence (runtime / FLOPs):** The authors provided average CPU/GPU runtime numbers for representative models, partially addressing the request for speed–accuracy evidence.
- **Lack of qualitative results:** The authors stated that visual comparisons were added to better illustrate qualitative differences.
- **Mechanistic clarification:** The authors expanded the motivation and added additional discussion/experiments (e.g., perturbation-related observations) to argue that SAM helps preserve edge/sharpness information.

**Remaining / outstanding concerns (key reasons for rejection):**
- **Marginal practical gains:** Across several settings, improvements remain small and do not clearly demonstrate a meaningful advance over strong baselines.
- **Insufficient “why” analysis:** The rebuttal does not fully resolve the request for a convincing explanation linking the sharpness-aware formulation to improved texture/detail recovery; deeper diagnostics (robustness/generalization evidence) remain limited.
- **Limited scope of validation:** Evaluation is still largely within linear inverse problems; claims about robustness/generalization to unseen degradations are not convincingly demonstrated.
- **Incremental contribution concern:** One reviewer maintained that the idea is interesting but incremental, and the current evidence does not justify acceptance.

**Reviewer Scores:**

- **Wr7b:** 4 (marginally below threshold) → likely **4**.
- **WntY:** 4 (marginally below threshold) → likely **4** .
- **MvHm:** 2 (reject) → likely **2** (possibly **4**).

---

### Decision · Program_Chairs · 2026-01-26

Reject